# Federated Continual Learning to Detect Accounting Anomalies in Financial Auditing

**Marco Schreyer**[1,2]    **Hamed Hemati**[1]    **Damian Borth**[1]    **Miklos A. Vasarhelyi**[2]

[1]University of St.Gallen, Switzerland    [2]Rutgers, The State University of New Jersey, USA

`{firstname.lastname}@unisg.ch, {firstname.lastname}@rutgers.edu`

## Abstract

The *International Standards on Auditing* require auditors to collect reasonable assurance that financial statements are free of material misstatement. At the same time, a central objective of *Continuous Assurance* is the 'real-time' assessment of digital accounting journal entries. Recently, driven by the advances in artificial intelligence, *Deep Learning* techniques have emerged in financial auditing to examine vast quantities of accounting data. However, learning highly adaptive audit models in decentralized and dynamic settings remains challenging. It requires the study of data distribution shifts over multiple clients and time periods. In this work, we propose a *Federated Continual Learning* framework enabling auditors to learn audit models from decentral clients continuously. We evaluate the framework's ability to detect accounting anomalies in common scenarios of organizational activity. Our empirical results, using real-world datasets and combined federated-continual learning strategies, demonstrate the learned model's ability to detect anomalies in audit settings of data distribution shifts.

## 1 Introduction

The *International Standards in Auditing (ISA)* demand auditors to collect reasonable assurance that financial statements are free from material misstatements, whether caused by error or fraud [58, 27]. The term *fraud* refers to *'the abuse of one's occupation for personal enrichment through the deliberate misuse of an organisation's resources or assets'* [80]. According to the *Association of Certified Fraud Examiners (ACFE)*, organisations lose 5% of their annual revenues due to fraud [1].[1] The ACFE highlights that respondents experienced a median loss of USD 100K in the first 7-12 months after a fraud scheme began. The timely detection of fraud and abuse is critical since the longer a scheme remains undetected, the more severe its financial, reputational, and organisational impact [54].

The digital transformation of the last decade has fundamentally changed the nature, recording, and volume of audit evidence [83]. Nowadays, organizations record vast quantities of digital accounting records, referred to as *Journal Entries (JEs)*, in *Enterprise Resource Planning (ERP)* systems [21]. For the audit profession, this unprecedented exposure to large volumes of accounting records offers new opportunities to obtain audit-relevant insights [2]. Lately, accounting firms also foster the development of *Deep Learning* (DL) capabilities [38] to learn advanced models to audit digital journal entry data. To augment a human auditor's capabilities, DL models are applied in various audit tasks, such as accounting anomaly detection [68, 87, 50] illustrated in Fig. 1, audit sampling [65, 62] or notes analysis [71, 55]. However, learning DL-enabled models in auditing is still in its infancy, exhibiting two main limitations. First, most of today's audit models are trained from

---

[1]The ACFE study encompasses an analysis of 2,110 cases of occupational fraud surveyed between January 2020 and September 2021 in 133 countries.

Workshop on Federated Learning: Recent Advances and New Challenges, held in Conjunction with the Conference on Neural Information Processing Systems (NeurIPS) 2022 (FL-NeurIPS'22).

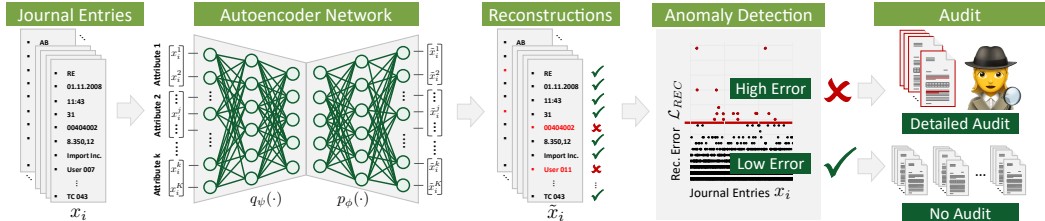

Figure 1: Overview autoencoder (AEN) based accounting anomaly detection setting [64, 68]. Journal entries corresponding to a high reconstruction error are selected for detailed audit procedures.

scratch on stationary client data of the in-scope audit period, *e.g.,* a financial quarter or year [72]. Disregarding that organizations operate in environments in which activities rapidly and dynamically evolve [78, 77, 24]. New business processes, models, or departments are constantly introduced, while current ones are redesigned or discontinued. Second, audit models are often trained centrally on a single client's data, *e.g.*, the organization 'in-scope' of the audit [24]. Although audit firms often audit multiple organizations operating in the same industry [26]. Such 'peer audit clients' [37] are affected by similar economic and societal factors, *e.g.*, supply-chains, market cycles [10].

In 1995 Thrun *et al.* proposed a machine learning setting in which a model incrementally learns from a stream of experiences [75]. The underlying idea of *Continual Learning (CL)* refers to a progressive model adaption; once new information becomes available [52]. In 2017 McMahan proposed *Federated Learning (FL)*, a learning setting enabling distributed clients to collaboratively train models under the orchestration of a central server [46]. A key idea of FL is creating learning synergies while preserving data privacy [17]. Ultimately, the *Federated Continual Learning (FCL)* of highly adaptive audit models can be viewed as a central objective of *Continuous Assurance* [78, 77]. However, learning such models in dynamic and decentralised audit settings remains challenging due to the *stability-plasticity* dilemma [22]. The FCL setting requires to cope with two types of data *distribution shifts* [20, 9, 30], as described in the following:

- A temporal distribution shift between time step $t_0$ and $t_1$ denotes a change in distributions $p_{t_0}(X) \neq p_{t_1}(X)$. In the CL setting, such shifts potentially result in the loss of a model's ability to perform on $p_{t_0}(X)$ after learning from $p_{t_1}(X)$ referred to as *catastrophic forgetting* [36].
- A client distribution shift between client $\omega_0$ and $\omega_1$ denotes a divergence of the distributions $p_{\omega_0}(X) \neq p_{\omega_1}(X)$. In the FL setting, such shifts cause the divergence of client models that eventually result in non-convergence of the server model referred to as *model interference* [32].

In this work, we investigate the practical application of FCL in financial auditing to improve the assurance of financial statements. In summary, we present the following contributions:

- We propose a novel federated continual learning framework that enables auditors to incrementally learn industry-specific models from distributed data of multiple audit clients.
- We demonstrate that the framework enables audit models to retain previously learned knowledge that is still relevant and replace obsolete knowledge with novel information.
- We evaluate scenarios of common organizational activity patterns using real-world datasets to illustrate the setting's benefit in detecting accounting anomalies.

Ultimately, we view FCL as a promising, under-explored learning setting in contemporary auditing. The remainder of this work is structured as follows: In Section 2, we provide an overview of related work. Section 3 follows with a description of the proposed framework to learn federated and privacy-preserving models from vast quantities of JE data. The experimental setup and results are outlined in Section 4 and Section 5. In Section 6, the work concludes with a summary.

## 2 Related Work

The application of ML in financial audits triggered a sizable body of research by academia [2, 72] and practitioners [73, 16]. This section presents our literature study focusing on the (i) unsupervised, (ii) federated, and (iii) continual learning of accounting data representations.

**Representation Learning** denotes a machine learning technique that allows systems to discover relevant data features to solve a given task [6]. Nowadays, most ML methods used in financial audits depend on 'human' engineered data representations [13]. Such techniques encompass, Naive Bayes

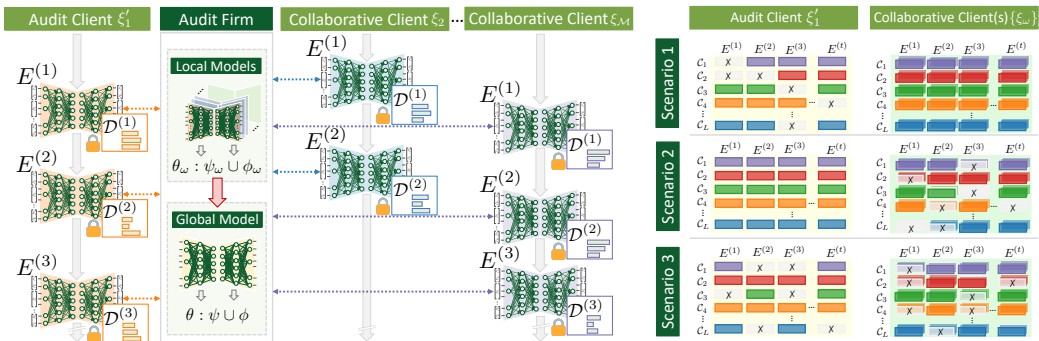

Figure 2: Schematic overview of the proposed *Federated Continual Learning (FCL)* setting [82] (left) to detect accounting anomalies and simulated organisational activity scenarios (right).

classification [5], network analysis [45], univariate and multivariate attribute analysis [3], cluster analysis [74], transaction log mining [33], or business process mining [29, 81]. With the advent of *Deep Learning* (DL), representation learning techniques have emerged in financial audits [72, 49, 55]. Nowadays, the application of DL in auditing JEs encompasses autoencoder neural networks [68], adversarial autoencoders [66], or variational autoencoders [87]. Lately, self-supervised learning techniques have been proposed to learn representations for multiple audit tasks [62].

**Continual Learning (CL)** describes a setting where a model continuously trains on a sequence of experiences [75]. The challenge of forgetting in model learning has been studied extensively [19, 15, 76, 40]. Rehearsal techniques replay samples from past experiences [59, 28, 12] to conduct knowledge distillation [56], while generative rehearsal techniques replay synthesized samples [69]. Recent methods use gradients of previous tasks to mitigate gradient interference [44, 11]. Regularisation techniques consolidate previously acquired knowledge and restrict parameter updates in model optimization. *LwF* [42], uses knowledge distillation to regularize parameter updates. *EWC* [36] directly regularises the parameters based on their previous task importance. In *SI* [85], regularization is applied in a separate parameter optimization step. Dynamic architecture techniques prevent forgetting by parameter reuse or increase [60, 48] or. Lately, CL has been deployed in a variety of application scenarios, such as healthcare [39], machine translation [4], and financial auditing [24].

**Federated Learning (FL)** enables entities to collaboratively train ML models under the orchestration of a central trusted entity under differential privacy [31, 17]. *FedAvg* [46] aggregates the client's model parameters by computing a weighted average based on the number of training samples. *PATE* [51] aggregates knowledge transferred from a teacher model that is trained on separated data of student models. *FedAdagrad*, *FedYogi*, and *FedAdam* [57] use adaptive optimizer variations to improve FL convergence. *FedProx* [41] trains the local with a proximal term that restricts the updates to the central model. *FedCurv* [70] aims to minimise the model dispersity across clients by adopting a modified version of EWC [36]. Recent works [84, 79] also introduce Bayesian non-parametric aggregation policies. Nowadays, FL learning is applied in sensitive application scenarios, such as healthcare [14], financial risk modelling [86], and financial auditing [63].

**Federated Continual Learning (FCL)** addresses the challenge of distributed distribution shifts over time and has only been preliminary studied [9]. *SCAFFOLD* [32] applies variance reduction in the local updates to correct for a client's concept drifts. *FedWeIT* [82] decompose a model into global shared and sparse task-specific parameters to prevent inter-client interference. *CDA-FedAvg* [9] extends FedAVG by distribution-based drift detection and a long-short term memory for rehearsal. *VER* [53] conducts an additional server-side rehearsal-based training using a VAE's [35] embedding statistics or actual embeddings. *FedDrift* [30] uses local drift detection and hierarchical clustering to learn multiple global concept models. To the best of our knowledge, this work presents the first step towards the federated continual learning of highly adaptive audit models in financial auditing.

## 3 Methodology

We consider an unsupervised audit learning setting where a tabular dataset $\mathcal{D}$ denotes a population of $i = 1, 2, ..., I$ JEs. Each entry $x_i = \{x_i^1, x_i^2, ..., x_i^J; x_i^1, x_i^2, ..., x_i^K\}$, consists of $j = 1, 2, ..., J$ categorical accounting attributes and $k = 1, 2, ..., K$ numerical attributes. The individual attributes encompass the journal entry details, such as posting date, amount, or general ledger. Each JE is generated

by a particular organizational activity $c_\ell$, *e.g.*, an organizational unit, entity or business process. Furthermore, $\mathcal{C} = \{c_1, c_2, ..., c_L\}$ denotes the set of all $\ell = 1, 2, ..., L$ activities. In our audit setting, we distinguish two types of 'anomalous' JEs [8] that auditors aim to detect [68, 87, 50]:

- **Global Anomalies** correspond to entries that exhibit unusual or rare individual attribute values, *e.g.*, rarely used ledgers or unusual posting times. Such anomalies often correspond to unintentional mistakes, are comparably simple to detect, and possess a high error risk.
- **Local Anomalies** correspond to entries that exhibit unusual attribute value correlations, *e.g.*, rare co-occurrences of ledgers and posting types. Such anomalies might correspond to intentional deviations, are comparably difficult to detect, and possess a high fraud risk.

Inspired by a human auditor's learning process, we introduce an audit framework comprised of three interacting learning settings, namely (i) *representation*, (ii) *continual*, and (iii) *federated* learning. Figure 2 illustrates the interactions of the settings described in the following.

First, in the **representation learning setting** *Autoencoder Networks (AENs)* [25] are trained to learn a comprehensive model $f_\theta$ of a given accounting data distribution $p(D = \{x_1, x_2, ..., x_I\})$. In general, the AEN architecture comprises two non-linear functions, usually neural networks referred to as *encoder* and *decoder*. The encoder $q_\psi(\cdot)$, with parameters $\psi$, learns a representation $z_i \in \mathcal{R}^{d_1}$ of a given input $x_i \in \mathcal{R}^{d_2}$, where $d_1 < d_2$. In an attempt to achieve $x_i \approx \tilde{x}_i$, the decoder $p_\phi(\cdot)$, with parameters $\phi$, learns a reconstruction $\tilde{x}_i \in \mathcal{R}^k$ of the original input. Throughout the training process, the AEN model parameters $\theta \colon \psi \cup \phi$ are optimized $\forall x_i \in \mathcal{D}$, as defined by [23]:

$$\theta^* \leftarrow \arg\min_{\psi,\phi} \|x_i - p_\phi(q_\psi(x_i))\| \, , \tag{1}$$

where $\theta^*$ denotes the optimal model parameters and $x_i$ a single JE. Upon successful training, the learning success is quantified by the model's reconstruction error $\mathcal{L}_{Rec}(f_\theta, \mathcal{D}, \tilde{\mathcal{D}})$ given $\mathcal{D}$ and its reconstruction $\tilde{\mathcal{D}}$. In our audit setting, similar to Hawkins *et al.* [23], we interpret an individual JEs reconstruction error magnitude as its deviation from regular posting patterns. JEs that correspond to a high $\mathcal{L}_{Rec}$ magnitude are selected for a detailed audit [68, 61] as shown in Fig. 1.

Second, in the **continual learning setting**, an AEN model $f_\theta$ learns from accounting data $\mathcal{D}$ that is observed as a stream of $M$ disjoint experiences $\{E^{(t)}\}_{t=1}^{\mathcal{T}}$ [75]. The data of the $t$-th experience $\mathcal{D}^{(t)}$ encompasses data instances of $\ell$ different activities $\{\mathcal{C}_\ell^{(t)}\}_{\ell=1}^L$ and an individual activity $\mathcal{C}_\ell^{(t)}$ corresponds to $\rho$ data observations $\{x_{\ell,s}^{(t)}\}_{s=1}^\rho$. With progressing experiences, the parameters $\theta^{(t)}$ of a given AEN model are then continuously optimized, as defined by [19]:

$$\theta^{*(t)} \leftarrow \arg\min_\theta \sum_{\ell=1}^L \mathcal{L}_{Rec}(f_\theta^{(t)}, \{x_{\ell,s}^{(t)}, \hat{x}_{\ell,s}^{(t)}\}_{s=1}^\rho | \theta_{init} = \theta^{*(t-1)}) \, , \tag{2}$$

where $\theta^{*(t-1)}$ denotes the optimal parameters of the previous experience $E^{(t-1)}$. In our audit setting each experience dataset $\mathcal{D}^{(t)}$ exhibits JEs of $L$ activities $\{\mathcal{C}_1^{(t)}, \mathcal{C}_2^{(t)}, ..., \mathcal{C}_L^{(t)}\}$, where in general $\mathcal{D}^{(t_1)} \neq \mathcal{D}^{(t_2)}$ given $t_1 \neq t_2$. Throughout the learning process, the distinct datasets $\mathcal{D}^{(t)}$ may be non-iid, and a model has only access to the data of an activity $\mathcal{C}_\ell^{(t)} \in \mathcal{D}^{(t)}$ during the time of the $t$-th experience. Although, a single organizational activity $\mathcal{C}_\ell$ might generate JEs in multiple experiences.

Third, in the **federated learning setting**, a central AEN model $f_\theta^{(t)}$ is collaboratively learned by $\mathcal{M}$ decentral clients $\{\xi_\omega\}_{\omega=1}^{\mathcal{M}}$ and coordinated by a server [31]. In each experience $E^{(t)}$, a single client $\xi_\omega$ exhibits access only to its private data subset $\mathcal{D}_\omega^{(t)} \in \mathcal{D}^{(t)}$. To initiate the learning, a synchronous model optimisation protocol is established that proceeds in rounds $r = 1, 2, ..., R$. Each round, the server broadcasts its central AEN model $f_\theta^{(t,r)}$ to a selection of available clients. Subsequently, each client conducts a decentral training of the central model on its data subset $\mathcal{D}_\omega^{(t)}$. Upon training completion, the clients send their updated models $f_{\theta,\omega}^{(t,r+1)}$ to the server. The server then aggregates the parameters of the decentral models to create an updated central AEN model $f_\theta^{(t,r+1)}$, as defined by [46]:

$$\theta^{(t,r+1)} \leftarrow \frac{1}{|\mathcal{D}^{(t)}|} \sum_{\omega=1}^{\mathcal{M}} |\mathcal{D}_\omega^{(t)}| \theta_\omega^{(t,r+1)} \, , \tag{3}$$

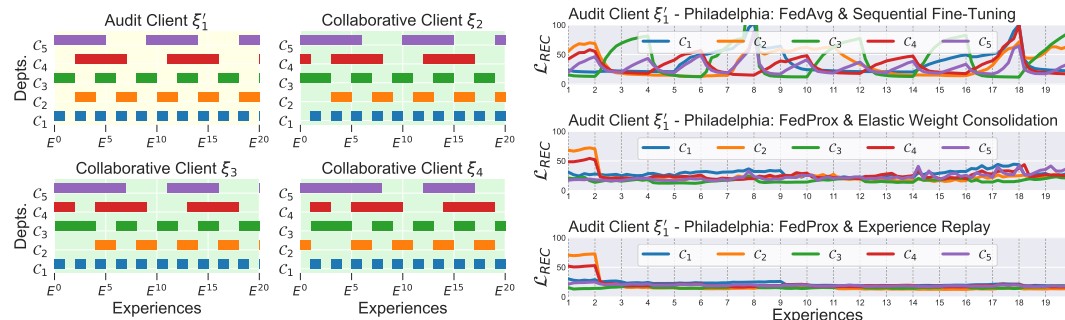

Figure 3: Example **[Scenario 3]** audit client $\xi_1'$ and collaborating clients $\{\xi_\omega\}_{\omega=2}^{\mathcal{M}}$ configuration (left). Activity (city department) reconstruction errors of $\xi_1'$ per strategy and progressing experiences (right).

where $|\mathcal{D}_\omega^{(t)}|$ denotes the number of data observations privately held by $\xi_\omega$. In our audit setting, a central audit firm coordinates the learning as illustrated in Fig. 2. We distinguish two categories of decentral FCL clients. First, an audit client $\xi_1'$ 'in-scope' or subject of the audit. Second, a set of collaborating 'peer' or collaborative clients $\{\xi_\omega\}_{\omega=2}^{\mathcal{M}}$ contributing to the learning of a central audit model. In each experience, the central model is applied to detect anomalies in the audit client's data.

## 4  Experimental Setup

In this section, we describe the (i) dataset, (ii) learning setting, and (iii) experimental details to detect accounting anomalies in an FCL setting. We provide additional experimental details in the appendix.

**City Payments Datasets:** To evaluate the FCL capabilities and allow for reproducibility, we use three publicly available datasets of real-world payment data. The datasets exhibit high similarity to ERP accounting data, *e.g.*, typical manual payments or payments runs (SAP T-Codes: F-35, F-110):

- The *City of Philadelphia* payments:[2] 238,894 payments of 58 city departments in 2017.
- The *City of Chicago* payments:[3] 108,478 payments of 54 city departments in 2021.
- The *City of York* payments:[4] 102,026 payments of 49 city departments in 2020-22.

Each dataset comprised of categorical and numerical attributes is described in the appendix.

**Federated Continual Learning:** We establish a systematic FCL setting encompassing (i) a central coordinating audit firm, (ii) a single decentral audit client $\xi_1'$, and (iii) three decentral collaborating clients $\{\xi_2, \xi_3, \xi_4\}$. The $\mathcal{M} = 4$ client models are learned in a federated setup, each over a different continual data stream $\{\mathcal{D}_\omega^{(t)}\}_{t=1}^{\mathcal{T}}$ of $\mathcal{T} = 20$ experiences [53]. Each experience, exhibits $R = 5$ learning rounds, while each round encompasses $\eta = 1,000$ training iterations. The data of an individual client experience $\mathcal{D}_\omega^{(t)}$, exhibits payments of $L = 5$ activities. A single experience activity $\mathcal{C}_{\omega,\ell} \in \mathcal{D}_\omega^{(t)}$ corresponds to $\rho = 1,000$ randomly sampled payments generated by a particular city department. The experience streams $\{\mathcal{D}_\omega^{(t)}\}_{t=1}^{\mathcal{T}}$ simulate data distribution shifts according to three common client activity scenarios. Each scenario corresponds to a different *sparse* client activity, where payment data of individual city department activities $\mathcal{C}_\ell$ is randomly observed. The distinct scenarios, illustrated in Fig 2, are described in the following:

- **[Scenario 1]** simulates a situation where the audit client $\xi_1'$ exhibits sparse payment activities, *e.g.*, due to discontinued or periodic business processes. The distribution shifts at $\xi_1'$ eventually yield catastrophic forgetting. The activities of the collaborating clients $\{\xi_\omega\}_{\omega=2}^{\mathcal{M}}$ remain constant.
- **[Scenario 2]** simulates a situation where collaborating clients $\{\xi_\omega\}_{\omega=2}^{\mathcal{M}}$ exhibits sparse payment activities, *e.g.*, due to a carve-out or merger. The distribution shifts at $\{\xi_\omega\}_{\omega=2}^{\mathcal{M}}$ eventually yield severe client model interference affecting $\xi_1'$. The activities of the audit client $\xi_1'$ remain constant.
- **[Scenario 3]** simulates a situation where an audit client $\xi_1'$ and the collaborating clients $\{\xi_\omega\}_{\omega=2}^{\mathcal{M}}$ exhibit sparse payment activities. The scenario simulates a common setting in which client activities opt-in and opt-out with progressing experiences causing model interference and forgetting.

---

[2] https://www.phila.gov/2019-03-29-philadelphias-initial-release-of-city-payments-data/

[3] https://data.cityofchicago.org/Administration-Finance/Payments/s4vu-giwb/

[4] https://data.yorkopendata.org/dataset/all-payments-to-suppliers/

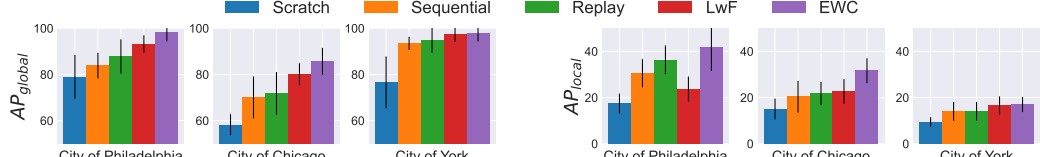

Figure 4: Audit client **[Scenario 1]** anomaly detection results comparing the CL techniques *Sequential Fine-Tuning*, *Replay* [56], *LwF* [42], and *EWC* [36] to mitigate catastrophic forgetting.

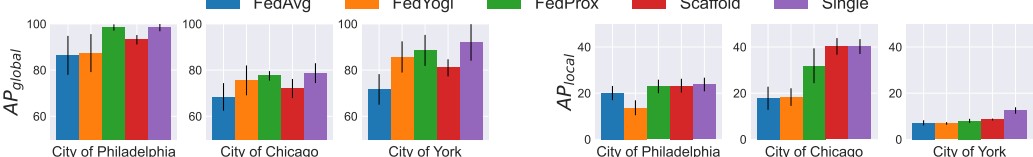

Figure 5: Audit client **[Scenario 2]** anomaly detection results comparing the FL techniques *FedAvg* [47], *FedYogi* [57], *FedProx* [41], and *Scaffold* [32] to mitigate the client's model interference.

**Accounting Anomaly Detection:** In each experience, we inject 20 global and 20 local anomalies[5] into each payment activity $\mathcal{C}_{\xi_1', \ell}$ of the audit client $\xi_1'$ ($\approx 4\%$ of each payment activity). To quantitatively assess the anomaly detection capability of the FCL audit setting, we determine the audit client's average precision ($AP$) of the detected anomalies over the stream of learning experiences $\{E^{(t)}\}_{t=1}^{\mathcal{T}}$ [61].

## 5   Experimental Results

In this section, we present the experimental results of scenarios 1-3. Our evaluation encompasses several federated (FL) and continual learning (CL) strategies. For each strategy, we report the audit client's $\xi_1'$ average global anomaly $AP_{global}$ and local anomaly $AP_{local}$ detection precision.

In Fig. 4, we report the **[Scenario 1]** audit client $\xi_1'$ anomaly detection results using *FedAvg* [47] model aggregation. For all datasets, the CL strategies, *Replay* [56], *LwF* [42], and *EWC* [36], outperform the from *Scratch* [24] learning and *Sequential* fine-tuning baselines. The results show the ability of the CL strategies ability to mitigate catastrophic forgetting in a sparse audit client setting.

In Fig. 5, we report the **[Scenario 2]** audit client $\xi_1'$ anomaly detection results conducting sequential model fine-tuning. For all datasets, the FL strategies, *FedYogi* [57], *FedProx* [41], and *Scaffold* [32], outperform the baseline of *FedAvg* [47]. In addition, the FL strategies mitigate model interference closing the gap towards *Single* [63] client learning in a sparse collaborating client setting.

Table 1: Audit client **[Scenario 3]** anomaly detection results comparing the combination of CL and FL techniques to mitigate catastrophic forgetting and the client's model interference.

| Learning Strategies | | City of Philadelphia | | City of Chicago | | City of York | |
|---|---|---|---|---|---|---|---|
| Federated | Continual | $AP_{global} \uparrow$ | $AP_{local} \uparrow$ | $AP_{global} \uparrow$ | $AP_{local} \uparrow$ | $AP_{global} \uparrow$ | $AP_{local} \uparrow$ |
| FedAvg [47] | Sequential | $74.03 \pm 9.51$ | $13.11 \pm 7.16$ | $48.16 \pm 8.41$ | $22.51 \pm 5.53$ | $87.37 \pm 8.74$ | $10.67 \pm 3.46$ |
| FedProx [41] | Replay [59] | $74.11 \pm 9.34$ | $20.21 \pm 0.00$ | $68.37 \pm 8.31$ | $22.90 \pm 5.46$ | $84.37 \pm 8.74$ | $10.77 \pm 2.23$ |
| FedProx [41] | LwF [42] | $\mathbf{99.84 \pm 2.97}$ | $18.02 \pm 3.95$ | $\mathbf{76.18 \pm 5.81}$ | $23.79 \pm 3.88$ | $\mathbf{88.23 \pm 6.16}$ | $10.79 \pm 3.30$ |
| FedProx [41] | EWC [36] | $96.39 \pm 3.58$ | $22.15 \pm 3.11$ | $64.30 \pm 3.11$ | $24.58 \pm 5.42$ | $87.67 \pm 3.11$ | $11.56 \pm 3.13$ |
| Scaffold [32] | Sequential | $93.39 \pm 4.23$ | $\mathbf{22.88 \pm 3.14}$ | $71.60 \pm 4.80$ | $\mathbf{26.66 \pm 7.22}$ | $74.34 \pm 8.77$ | $\mathbf{11.67 \pm 3.46}$ |

*Variances originate from training using five distinct random seeds of model parameter initialisation and anomaly sampling.

In Tab. 1, we report the **[Scenario 3]** audit client $\xi_1'$ anomaly detection results of combining FL and CL strategies. The strategies yield a superior precision for both anomaly classes compared to the *FedAvg-Sequential* baseline. The obtained results suggest that certain strategies are advantageous in detecting particular anomaly classes. For smaller (larger) AEN models, we observe that *FedProx* (*Scaffold*) learning successfully regulates dissimilar model convergence. The results provide initial empirical evidence, as illustrated in Fig. 3, that the strategies mitigate distributed-continual distribution shifts in a 'real-world' audit setting.

---

[5]To sample both anomaly classes, we use the *Faker* library [18]: `https://github.com/joke2k/faker`.

## 6  Conclusion

In this work, we proposed financial audits as a novel application area of *Federated Continual Learning* (FCL). We demonstrated the benefits of such a learning setting by detecting accounting anomalies under different data distribution shifts usually observable in organizational activities. We believe that FCL will enable the audit profession to enhance its assurance services and thereby contributes to the integrity of financial markets. In future work, we aim to investigate (i) novel learning strategies, (ii) potential attack scenarios, and (iii) the addition of differential privacy.

## Acknowledgments and Disclosure of Funding

The research conducted by Marco Schreyer was funded by the Mobi.Doc mobility grant of the University of St.Gallen (HSG) under project no. 1031606.

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

# A Appendix

In the appendix, we provide additional details of the datasets, data preprocessing, federated continual learning scenarios, and experimental setup applied to detect accounting anomalies in an FCL setting.

## A.1 Datasets and Data Preprocessing:

In our experiments, we use three publicly available datasets of real-world financial city payment data. The datasets exhibit high similarity to ERP accounting data, *e.g.*, typical manual payments (SAP T-Code: F-53) or payment runs (SAP T-Code: F-110):

- The *City of Philadelphia* payments[6] encompass a total of 238,894 payments generated by 58 city departments in 2017. Each payment exhibits 10 categorical and 1 numerical attribute(s).
- The *City of Chicago* payments[7] encompass a total of 108,478 payments generated by 54 distinct city departments in 2021. Each payment exhibits 6 categorical and 1 numerical attribute(s).
- The *City of York* payments[8] encompass a total of 102,026 payments generated by 49 city departments in 2020-22. Each payment exhibits 11 categorical and 2 numerical attribute(s).

For each dataset $\mathcal{D}$, we pre-process the original payment line-item attributes to (i) remove semantically redundant attributes and (ii) obtain an encoded representation of each payment. The following pre-processing is applied to the distinct payment attributes of the datasets:

- For each categorical attribute in $\mathcal{D}$, the attribute values $x_i^k$ are converted into one-hot numerical tuples of bits $\hat{x}_i^k \in \{0,1\}^\upsilon$, where $\upsilon$ denotes the number of unique attribute values in $x^k$.
- For each numerical attribute in $\mathcal{D}$, the attribute values $x_i^l$ are scaled, according to $scale(x_i^l) = (x_i^l - min(x^l)/(max(x^l) - min(x^l))$, based on the $min$ and $max$ attribute value in $x^l$, where

In the following, the pre-processed journal entries are denoted as $\hat{x} \in \hat{X}$ and the corresponding individual attributes as $\hat{x}_i = \{\hat{x}_i^1, \hat{x}_i^2, ..., \hat{x}_i^J; \hat{x}_i^1, \hat{x}_i^2, ..., \hat{x}_i^K\}$, where the index $j = 1, 2, ..., J$ denotes the categorical accounting attributes. The index $k = 1, 2, ..., K$ denotes the numerical attributes.

## A.2 Federated Continual Learning (FCL) Scenarios

For each of the $\mathcal{M}$ federated clients $\{\xi_\omega\}_{\omega=2}^{\mathcal{M}}$, we create a different city payment data stream encompassing $\mathcal{T} = 20$ experiences. The data stream of single federated client $\xi_\omega$ is defined by $\mathcal{D}_\omega = \{\mathcal{D}_\omega^{(1)}, \mathcal{D}_\omega^{(2)}, ..., \mathcal{D}_\omega^{(20)}\}$. For each single experience dataset $\mathcal{D}_\omega^{(t)}$, we randomly sample payments from $L = 5$ distinct payment activities $\mathcal{C}_\ell$. Thereby, each payment activity corresponds to a different city department. In our experiments, we sampled from the following city departments to create the data streams for each dataset respectively:

- The selected *City of Philadelphia* departments are (i) '42 Commerce', (ii) '52 Free Library', (iii) '10 Managing Director', (iv) '11 Police', and (v) '14 Health'.
- The selected *City of Chicago* departments are (i) 'Dept. of Family and Suppport Services', (ii) 'Dept. of Aviation', (iii) 'Chicago Department of Transportation', (iv) 'Department of Health', and (v) 'Department of Water Management'.
- The selected *City of York* departments are (i) 'Adult Social Care', (ii) 'Economy Regeneration and Housing', (iii) 'Housing and Community Safety', (iv) 'Transport Highways and Environ.', and (v) 'School Funding and Assets'.

For each dataset, the selected departments exhibit a high payment posting activity. Furthermore, the departments correspond to different municipal duties. In total, we created three FCL activity scenarios as described in Sec. 4 and illustrated in Fig. 2. For each client payment activity $\mathcal{C}_{\omega,\ell}^{(t)}$, the payments are sampled according to three scenario configurations, denoted as [Scenario 1-3] in the following. Figure 6 illustrates the bar charts of each sampling configuration and FCL client. For each client configuration, a bar observable for client activity $\mathcal{C}_\ell$ and experience $E^{(t)}$ denotes that $\rho = 1,000$ city payments are sampled in the corresponding scenario.

---

[6]https://www.phila.gov/2019-03-29-philadelphias-initial-release-of-city-payments-data/

[7]https://data.cityofchicago.org/Administration-Finance/Payments/s4vu-giwb/

[8]https://data.yorkopendata.org/dataset/all-payments-to-suppliers/

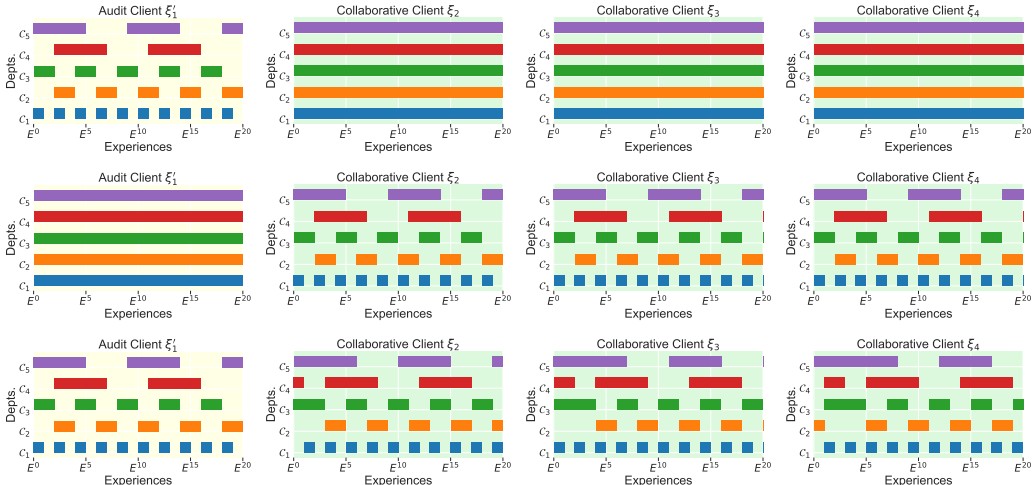

Figure 6: Experimental **[Scenario 1-3]** configurations of the evaluated FCL settings. The distinct configurations illustrate the payment activity per department at each of the $\mathcal{M} = 4$ federated clients: **[Scenario 1]** top-row, **[Scenario 2]** middle-row, and **[Scenario 3]** bottom-row.

### A.3 Representation Learning Setting Details

We use a symmetrical encoder $q_\psi$ and decoder $p_\phi$ architecture, in all our experiments. Furthermore, we applied two architectural setups, each designed to detect a different activity of injected anomalies in both datasets. A 'shallow' architecture, as shown in Tab. 2, is designed to detect global anomalies and a 'deep' architecture, as shown in Tab. 3, is designed to detect local anomalies.

Table 2: Number of neurons per layer $\nu_i$ of the encoder $q_\psi$ and decoder $p_\phi$ network that constitute the AEN architecture to detect *global anomalies* used in our experiments.

| Layer $\nu_i$ | $i = 1$ | 2 | 3 | 4 | 5 | 6 | 7 | 8 |
|---|---|---|---|---|---|---|---|---|
| $q_\psi(z\|\hat{x})$ | $\|\hat{x}\|$ | 128 | 64 | 32 | 16 | 8 | 4 | 2 |
| $p_\phi(\tilde{x}\|z)$ | 2 | 4 | 8 | 16 | 32 | 64 | 128 | $\|\hat{x}\|$ |

Table 3: Number of neurons per layer $\nu_i$ of the encoder $q_\psi$ and decoder $p_\phi$ network that constitute the AEN architecture to detect *local anomalies* used in our experiments.

| Layer $\nu_i$ | $i = 1$ | 2 | 3 | 4 | 5 | 6 | 7 | 8 | 9 | 10 | 11 | 12 |
|---|---|---|---|---|---|---|---|---|---|---|---|---|
| $q_\psi(z\|\hat{x})$ | $\|\hat{x}\|$ | 2048 | 1024 | 512 | 256 | 128 | 64 | 32 | 16 | 8 | 4 | 2 |
| $p_\phi(\tilde{x}\|z)$ | 2 | 4 | 8 | 16 | 32 | 64 | 128 | 256 | 512 | 1024 | 2048 | $\|\hat{x}\|$ |

In both architectures, we apply Leaky-ReLU non-linear activations with a scaling factor $\alpha = 0.4$ except in the encoder's bottleneck and decoder's final layer, which comprise Tanh activations. We use a batch size of $\gamma = 16$ journal entries per training iteration $\eta$, apply Adam optimization [34] with $\beta_1 = 0.9$, $\beta_2 = 0.999$, and early stopping once the reconstruction loss $\mathcal{L}_{Rec}$ converges. Given an encoded journal entry $\hat{x}_i$ and its reconstruction $\tilde{x}_i = p_\phi(q_\psi(x_i))$, we compute a combined loss of a binary cross-entropy error $\mathcal{L}_{BCE}$ loss and a mean squared error $\mathcal{L}_{MSE}$ loss, as defined by [67]:

$$\mathcal{L}_{Rec}(f_\theta, \{x_i, \tilde{x}_i\}) = \vartheta \sum_{j=1}^{J} \mathcal{L}_{BCE}\left(f_\theta, \{\hat{x}_i^j, \tilde{x}_i^j\}\right) + (1 - \vartheta)\sum_{k=1}^{K} \mathcal{L}_{MSE}\left(f_\theta, \{\hat{x}_i^k, \tilde{x}_i^k\}\right), \quad (4)$$

where $J$ denotes the number of categorical attributes, $K$ the number of numerical attributes, and $\vartheta$ balances the categorical and numerical attribute losses. For each categorical attribute $\hat{x}_i^j$, we compute the $\mathcal{L}_{BCE}$, as defined by:

$$\mathcal{L}_{BCE}(\hat{x}_i^j, \tilde{x}_i^j) = \frac{1}{\Upsilon} \sum_{v=1}^{\Upsilon} \tilde{x}_{i,v}^j \log(\hat{x}_{i,v}^j) + (1 - \tilde{x}_{i,v}^j)\log(1 - \hat{x}_{i,v}^j), \quad (5)$$

where $v = 1, 2, ..., \Upsilon$ corresponds to the number of one-hot encoded attribute dimensions of a given categorical attribute. For each numerical attribute $\hat{x}_i^k$, we compute the $\mathcal{L}_{MSE}$, as defined by:

$$\mathcal{L}_{\psi,\phi}^{MSE}(\hat{x}_i^k, \tilde{x}_i^k) = (\hat{x}_i^k - \tilde{x}_i^k)^2 . \tag{6}$$

We balance both loss terms in $\mathcal{L}_{MSE}$ to account for the high number of categorical attributes in each city payment dataset and set $\vartheta = \frac{2}{3}$ in all experiments.

## A.4 Continual Learning Setting Details

We set the number of CL experiences $\mathcal{T} = 20$ in all our experiments. In *Experience Replay* (ER) [59] learning, we set the replay buffer size $N_B = 1,000$ samples, which seemed sufficient in all datasets. The buffer keeps a stratified sample of all departments observed until the current experience. For *Elastic Weight Consolidation* (EWC) [36], we found that $\alpha_{EWC} = 500$ preserves an optimal degree of plasticity. For *Learning without Forgetting* (LwF) [42] we determined that a $\alpha_{LwF} = 1.2$ yields a good knowledge distillation. We reset the optimizer parameters upon each experience to avoid past experience information transfer through the optimizer state. We adapted the different CL strategies from the algorithmic implementations of ER, LwF, and EWC of the Avalanche v0.2.1[9] library [43].

## A.5 Federated Learning Setting Details

The models are trained for $R = 5$ communication rounds per experience, while each round encompasses $\eta = 1,000$ training iterations. In each experiment, we set the number of participating clients to $\mathcal{M} = 4$ assuming a real-world peer audit client setting. In *FedProx* [41] learning, we found that setting the proximal term to $\mu = 1.2$ yields a good regularization of the local model updates. For the *SCAFFOLD* [32] algorithm, we use the Option II implementation proposed in [32] by re-using previously computed gradients to update the control variate $c_i^+$. In all our experiments we built upon the FedAvg strategy and the FL framework implementation of the Flower v0.19.0[10] library [7].

## A.6 Anomaly Detection Details

In each experience, we randomly inject 20 global and 20 local anomalies into each payment activity $\mathcal{C}_{\xi_1',\ell}$ of the in-scope audit client $\xi_1'$ ($\approx 4\%$ of each payment activity). To create both classes of anomalies, we use the Faker v14.2.0 [11] library [18] using five distinct random seed initialisation of the random anomaly sampling mechanism. To quantitatively assess the anomaly detection capability of the FCL audit setting, we determine the in-scope audit client's *Average Precision* ($AP$) over the sorted payments reconstruction errors. The $AP$ summarises the precision-recall curve as defined by:

$$AP(\mathcal{L}_{Rec}) = \sum_{i=1}^{N} (R_i(\mathcal{L}_{Rec}) - R_{i-1}(\mathcal{L}_{Rec}))P_i(\mathcal{L}_{Rec}) , \tag{7}$$

where $P_i(\mathcal{L}_{Rec}) = TP/(TP + FP)$ denotes the detection precision, and $R_i(\mathcal{L}_{Rec}) = TP/(TP + FN)$ denotes the detection recall of the i-*th* reconstruction error threshold.

---

[9] https://github.com/ContinualAI/avalanche

[10] https://github.com/adap/flower

[11] https://github.com/joke2k/faker

