# OpenReview forum: "Federated Continual Learning to Detect Accounting Anomalies in Financial Auditing"
_NeurIPS.cc/2022/Workshop/Federated_Learning — FL-NeurIPS 2022 Poster_

### Official Review · Reviewer_26NM · 2022-10-11
**Overall solid paper, could be better written**

This paper looks at a number of strategies for deep learning in financial auditing, where the challenge is to do anomaly detection against a globally shifting distribution. The paper compares the performance of a number of existing strategies on different datasets. Overall the paper is moderately interesting.

For further feedback, I would encourage the authors to more clearly indicate what the main findings are. While the paper is overall well written, the main message of the experimental section seems a little lost.

---

### Official Review · Reviewer_nMBP · 2022-10-16
**The paper addresses a relevant topic. However, the descriptions of the proposed method and the structure of proposed method are not well introduced.**

This paper proposes an FCL method to detect anomalies in financial auditing. The paper addresses a relevant topic. And the case studies are well presented. However, the descriptions of the proposed method and the structure of proposed method are not well introduced. Besides, detailed contributions in the paper are less described. Also, the presented theory is less, though there are some English inconsistencies that need to be corrected. Careful proofreading is necessary.
1.	Some mathematical symbols do not explain the meanings, and formulas do not introduce principles. Please use the prescriptive grammar and formula to give the explanation.
2.	The description of Fig. 2 and Table 1 are not enough, please give more descriptions.
3.	It would be better if the authors can show the pseudocode of the proposed method.
4.	The authors do not show how to select some parameters, please list the parameter settings.

---

### Author Response · Authors · 2022-11-29
**Final Version and Reviewers Response**

We are grateful for the reviewer's paper acceptance and valuable feedback.

In the final version's appendix, we added further details on the experimental setup. Furthermore, we enhanced the version's experimental results and conclusion to accommodate the reviewer's remarks.

We hope to have addressed the concerns and look forward to the workshop.

---

### Decision · Program_Chairs · 2022-10-20

Accept (Poster)